# Estimating uncertainty from feed-forward network based sensing using quasi-linear approximation

## Abstract

Artificial neural networks are increasingly integrated into both sensing hardware (e.g., "smart sensors") and dedicated decision-making circuits that operate on this information. As this technology is deployed in safety-critical environments (pedestrian-detection, power management, and flight-controls) it is critical to assess the real-time confidence of information built on these networks. However, while stand-alone confidence of sensing (e.g. object detection) neural networks are common, tools are much more limited for integrating such information into formal estimation of latent variables upstream of the sensor. To make this distinction clear, consider the common problem of target-tracking from a mobile camera. The geographic position of the target is a function of the camera position and orientation in addition to position within the image, whereas the neural network only reports confidence in pixel-space. Likewise, optimally leveraging an image-sequence requires consideration of uncertainty in the camera and target dynamics, as well as the sensing neural network. As we will demonstrate, fusing dynamical system models with large sensing networks presents a major computational challenge. Specifically, popular approaches such as first-order (Jacobian) linearization prove inaccurate, whereas nonlinear sampling-based approaches, while effective, are intractable for high-dimensional measurements such as images. In this work, we borrow an analytic approach from control engineering, quasilinear system approximation, to propagate the dynamics of environmental uncertainty through feedforward neural network architectures. The approximation enables direct Bayesian (i.e., Kalman-style) filtering to estimate latent variables, thus obviating the need for taxing sampling-based approaches. Thus, the proposed framework may enable real-time confidence estimation in high-dimensional network-based sensing deployments.

## 1 Introduction

Decision-making systems are increasingly reliant on sensors whose noise characteristics are opaque, owing to the integration of artificial neural networks (ANNs) into their design. These 'smart-sensors' offer powerful capability, but make it difficult to ascribe confidence to ensuing estimates by systems that must act on the sensor output information. Applying standard approaches in estimation and filtering theory is difficult due to the analytical difficulties caused by the NNs. Increasingly, these types of sensing implementations are deployed in safety-critical cyber-physical systems such as pedestrian-detection systems, power-plant control, and flight-controls. It is therefore important to be able to assess the confidence of outputs emanating from these networks in real-time applications. In this regard, it is increasingly standard to embed confidence as an output in stand-alone sensing implementations of neural networks (e.g., for object detection or tracking). However, when the goal is to determine uncertainty in dynamical processes upstream of the sensor, tools are more limited.

To illustrate this distinction, consider the problem of target-tracking using an aerial drone. The geographic position of the target cannot be determined from only its position in the image, since one must also know the position and orientation of the drone itself. However, a neural network detector in this setting might only report confidence in pixel space. Likewise, optimally leveraging an image-sequence requires consideration of uncertainty in the drone/target dynamics as well as

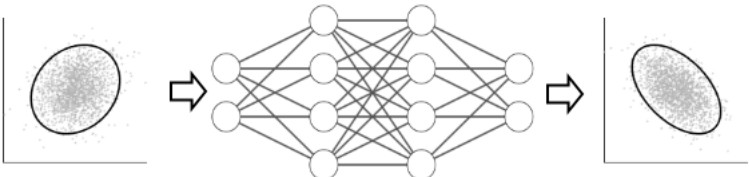

Figure 1: The input of a pre-trained feed-forward neural network is perturbed with Gaussian noises, and its uncertainty also undergoes a non-linear transformation, of which an analytical form is not available. We need a method that can estimate output covariance in real time in order to enable Kalman filtering for latent variable inference.

the sensing neural network. As we will demonstrate, fusing dynamical system models with large sensing networks presents a non-trivial challenge. Specifically, basic approaches such as first-order (Jacobian) linearization prove to be quite inaccurate, whereas nonlinear sampling-based approaches are currently intractable for high-dimensional measurements such as images.

To overcome this challenge, we need to analytically, if not solve, then estimate output uncertainty caused by input uncertainty, which requires inference of how the moments, and at minimum the mean and covariance, of a distribution propagate through every layer of a NN (Fig. 1). In this paper, we borrow an analytic approach from control engineering, the quasilinear approximation, to propagate the dynamics of environmental uncertainty through neural networks. The method of quasilinear approximation and stochastic linearization has been developed primarily in the context of feedback control systems (see, e.g., Ching et al. (2010), Socha (2007), Roberts & Spanos (2003), Elishakoff & Crandall (2017) for an overview). In particular, Kabamba et al. (2015) focuses on stochastic linearization, a quasilinear approximation approach, of univariate nonlinearity, and the multivariate nonlinearities case is discussed in Atalik & Utku (1976). Caughey proposes a similar approach, known as equivalent linearization Caughey (1963), which is often interchangeable with stochastic linearization. The existence and uniqueness of solution given by this method is discussed in Spanos & Iwan (1978). As suggested by Socha and Pawleta, there is a close relationship between stochastic linearization and statistical linearization Socha & Pawleta (2001).

**Related work**     Our research is also related to prior efforts in *uncertainty propagation*. Much of this work focuses on estimating mean and variance Ewert et al. (2014), Astudillo & da Silva Neto (2011), Gast & Roth (2018), Wang & Manning (2013) in NN settings, but often requires assumptions of output independence to enable calculations regarding propagation through layers. These approaches also neglect correlation between inputs, which can lead to significant errors, especially in multi-layer networks. For example, Gandhi et al. (2018) uses a linearization-based methodology to estimate uncertainty in NNs that shares similarity with our proposed approach, however it makes an assumption that inputs are uncorrelated which limits applicability. Shekhovtsov and Flach developed an analytic form of propagation through sigmoid and softmax layers Shekhovtsov & Flach (2019), but required logistic and Gumbel priors on noise. Perhaps the closest to what we propose here is Wang et al. (2016) in which Wang et. al. gave an analytic estimation of mean and variance of independent Gaussian noise propagating through a layer of sigmoidal and $\mathbf{tanh}$ transformations. However, that work is developed in probabilistic neural networks and the formulation is not easily applicable in a generic neural network setting.

**Contribution**     It is worth clarifying here that our goal is strictly in terms of estimating input-output uncertainty propagation. Though we are aware of potential applications of our method in Bayesian weight learning, autopilot, etc., the discussion of broader impact in those areas is out of the scope of this paper. Rather, we seek a means to estimate how noise injected at an input of a pre-trained network with fixed weights affects its outputs. We will show that our proposed method is effective in this regard, and enables us to efficiently close the loop between neural network sensing and physical systems with Bayesian (Kalman-style) Filtering. We validate this approach for target-tracking in a simulated environment.

## 2 METHODS

### 2.1 QUASILINEAR CONTROL (QLC) AND STOCHASTIC LINEARIZATION (SL)

The theory of QLC was developed to study nonlinear stochastic control systems Ching et al. (2010) by leveraging SL. SL is a technique for approximating a nonlinearity with an affine function whose gain and bias are based on statistical properties of nonlinearities input-output relationship Atalik & Utku (1976). Brahma and Ossareh formulated SL of both open-loop and feedback stochastic systems Brahma & Ossareh (2019): a piecewise differentiable nonlinear function $f(\cdot)$ can be approximated by $N^\top u_\circ(t) + M$, where $u_\circ = u(t) - \mathbb{E}[u(t)]$ was the zero-mean part of input $u(t)$. N, the *quasilinear gain*, M the *quasilinear bias*, were denoted by:

$$N = \mathbb{E}[\nabla f(u(t))] \qquad M = \mathbb{E}[f(u(t))] \tag{1}$$

Here, the expectations are taken with respect to the distribution of $u$, and hence the method is much more sensitive to the full shape of the function $f(\cdot)$ than the traditional Jacobian linearization performed locally at a single point.

Assuming Gaussian noises injected to input $u$, intermediate output vectors of each layer can be estimated by multivariate Gaussian random variables after stochastically linearizing the network. Since the intermediate output vectors of each layer can be approximated as Gaussian random vectors, the stochastically linearized neural network becomes a conditional Bayesian network Neal (1992), the expectation propagation through which has been discussed in Shekhovtsov & Flach (2019).

In our proposed framework, we will apply this methodology to approximate the nonlinearities in a neural network, and thus derive an analytic solution of propagated variance-covariance, which neither requires computationally demanding sampling, nor is subject to the 'curse of dimensionality'.

### 2.2 UNCERTAINTY PROPAGATION THROUGH **tanh** (AND ANY SIGMOIDAL) LAYERS

The output of a perceptron, a single computing unit of a **tanh** layer, is:

$$f(u) = \mathbf{tanh}(u + b) \quad \text{where} \quad u = w^\top x, \tag{2}$$

$x$ are input variables (e.g., pixel values of an image), $w$ is weight vector, and $b$ is bias of the perceptron. The layers of the perceptron together form an open-loop, nonlinear system. The objective is to estimate the variance and covariance of the output.

We assume the latent variables $x$ are injected with multivariate Gaussian noises $\xi$ (Eqn. 4). Note it is quite arduous to solve the integral $\int \mathbf{tanh}'(x + \xi + b)p(\xi)\mathrm{d}\xi$ analytically, so we substitute $f = \mathbf{tanh}(\cdot)$ with a sigmoidal function that is very similar but more tractable. Specifically, we propose

$$\mathbf{tanh}(u + b) \approx g(u) \triangleq \frac{1}{p}\sum_{j=1}^{p} \mathbf{erf}\left[\gamma_j(u + b)\right] \tag{3}$$

where $\{\gamma_1, ..., \gamma_p\}$ is a set of scaling factors in scalar values. Introducing this reformulation of the **tanh** function not only makes the marginalization possible (convolution of two Gaussian functions yields a Gaussian function), but also maintains a favorable error. In our experiments, we choose a set of two scaling parameters, $\{0.64, 1.13\}$, fitted by minimizing mean square error. In practice, one can add more parameters for even higher accuracy without adding computation time (depending on the computing resources), because the extra terms can be easily paralleled.

We are now ready to proceed with the main approximation. We assume that the input to the **tanh** layer, $x$ has uncertainty modeled by $\xi$, so that with a slight abuse of notation,

$$u = w^\top(x + \xi) \quad , \text{where } \xi \sim \mathcal{N}(0, \Sigma) \tag{4}$$

$$g(u) = \frac{2}{p\sqrt{\pi}}\sum_{j=1}^{p}\int_{0}^{\gamma_j(u+b)} \mathbf{exp}(-\nu^2)\mathrm{d}\nu \tag{5}$$

Then, denoting $z \triangleq w^\top x + b$, and $\varphi(\cdot)$ as the standard Gaussian probability density function, we derive the SL model of a **tanh** layer in (Eqn. 6-9) (see Appendix for detailed derivation). The output

follows a multivariate Gaussian transition probability density, because it is a linear function of $w^\top \xi$. The quasilinear gain $N$ is a Gaussian function of zero-noise input $w^\top x$ or $z$ if zero-centered, since it is the convolution of two Gaussian functions. We obtain the quasilinear bias by integrating the gain, giving a function of the Gaussian cumulative density function (i.e., **erf**). The superscript $k = 0, 1, 2, ...$ indexes each layer with $k = 0$ as the input layer. Recursively, we can approximate the mean and covariance matrix of every layer. Lastly, the SL model of layers with sigmoidal activations (e.g. logistic) can be obtained in a similar way.

$$\textbf{Gain} \quad N^{(k)} = \frac{2}{p} \sum_{j=1}^{p} \frac{1}{\hat{\sigma}_j^{(k)}} \odot \varphi\left(\frac{z^{(k)}}{\hat{\sigma}_j^{(k)}}\right) \qquad \textbf{Bias} \quad M^{(k)} = \frac{1}{p} \sum_{j=1}^{p} \textbf{erf}\left(\frac{z^{(k)}}{\sqrt{2\hat{\sigma}_j^{(k)^2}}}\right) \qquad (6)$$

$$\text{where} \quad \hat{\sigma}_j^{(k)^2} = \frac{2\gamma_j^2 \tilde{\sigma}^{(k)^2} + 1}{2\gamma_j^2} \quad \text{and} \quad \tilde{\sigma}^{(k)^2} = \textbf{diag}\left(w^{(k)^\top} \Sigma^{(k-1)} w^{(k)}\right) \qquad (7)$$

$$\textbf{Zero-Mean Input} \quad u_\circ^{(k)} = w^{(k)^\top} \xi^{(k-1)} \quad \textbf{Output} \quad y^{(k)} = N^{(k)} \odot u_\circ^{(k)} + M^{(k)} \qquad (8)$$

and the variance-covariance matrix of the output of any stochastically linearized layer is

$$\Sigma^{(k)} = \left(N^{(k)} N^{(k)^\top}\right) \odot \left(w^{(k)^\top} \Sigma^{(k-1)} w^{(k)}\right) \qquad (9)$$

## 2.3 UNCERTAINTY PROPAGATION THROUGH **softplus** LAYERS

A **softplus** activation is defined by

$$\textbf{softplus}(u + b) = \frac{1}{\beta} \textbf{log}(1 + e^{\beta(u+b)}) \qquad (10)$$

For simplicity, we choose $\beta = 1$ as default value. With the same notation of variables in the previous section, we propose the approximation

$$\frac{\mathrm{d}}{\mathrm{d}u} \textbf{softplus}(u + b) = \frac{1}{1 + e^{-(u+b)}} \approx \frac{1}{p} \sum_{j=1}^{p} \mathbf{\Phi}\left(\frac{\gamma_j}{\sqrt{2}}(u + b)\right) \qquad (11)$$

Here $\mathbf{\Phi}(\cdot)$ is the standard normal cdf. Then we can obtain the quasilinear gain $N$ by doing a convolution of a Gaussian cdf and a Gaussian pdf, and quasilinear bias $M$, integral of a Gaussian cdf (Eqn. 12-13) (see Appendix for detailed derivation). The SL model of **GELU** layers can also be derived in a similar way, because $\nabla \textbf{GELU}(x) = \mathbf{\Phi}(x) + x\varphi(x)$.

$$\textbf{Gain} \quad N^{(k)} = \frac{1}{2p} \sum_{j=1}^{p} \textbf{erfc}\left(-\frac{z^{(k)}}{\sqrt{2\hat{\sigma}_j^{(k)2}}}\right) \quad \textbf{Bias} \quad M^{(k)} = N^{(k)} \odot z^{(k)} + \frac{1}{p} \sum_{j=1}^{p} \hat{\sigma}_j^{(k)} \odot \varphi\left(\frac{z^{(k)}}{\hat{\sigma}_j^{(k)}}\right) \qquad (12)$$

$$\text{where} \quad \hat{\sigma}_j^{(k)^2} = \frac{\gamma_j^2 \tilde{\sigma}^{(k)^2} + 2}{\gamma_j^2} \quad \text{and} \quad \tilde{\sigma}^{(k)^2} = \textbf{diag}\left(w^{(k)^\top} \Sigma^{(k-1)} w^{(k)}\right) \qquad (13)$$

## 2.4 UNCERTAINTY PROPAGATION THROUGH **ReLU** (ANY ANY PIECE-WISE LINEAR) LAYERS

The derivation of SL model of **ReLU** layers is straightforward (Eqn. 14-15), and does not requires approximation (see Appendix for detailed derivation). The SL model of layers with any piece-wise linear activation function (e.g. leaky ReLU) can be obtained in a similar way.

$$\textbf{Gain} \quad N^{(k)} = \frac{1}{2} \left[1 + \textbf{erf}\left(\frac{z^{(k)}}{\sqrt{2\tilde{\sigma}^{(k)2}}}\right)\right] \quad \textbf{Bias} \quad M^{(k)} = N^{(k)} \odot z^{(k)} + \tilde{\sigma}^{(k)} \odot \varphi\left(-\frac{z^{(k)}}{\tilde{\sigma}^{(k)}}\right) \qquad (14)$$

$$\text{where} \quad \tilde{\sigma}^{(k)^2} = \textbf{diag}\left(w^{(k)^\top} \Sigma^{(k-1)} w^{(k)}\right) \qquad (15)$$

# 3 RESULTS

## 3.1 APPROXIMATION ACCURACY OF STOCHASTIC LINEARIZATION

We introduce SL to account for the influence of input covariance on the uncertainty propagation through non-linear transformations. We compare predicted output variance and mean by JL and SL (Fig. 2) with ground truth under different assumptions about input variance, and observe alignment between SL prediction and the true values. As expected, the underestimation of variance propagation by SL becomes more significant as input variance increases. JL, by taking first order Taylor expansion, is invariant to the input covariance in these simulations.

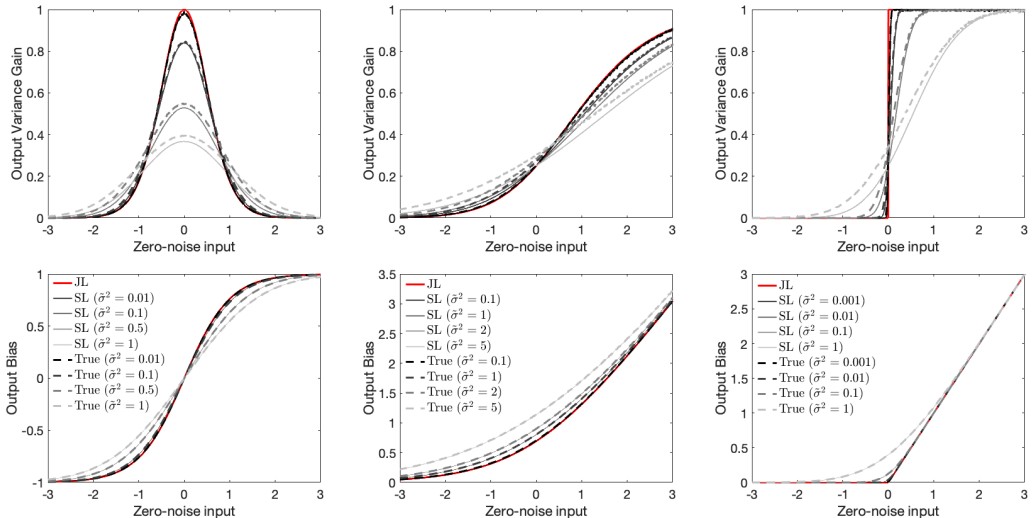

Figure 2: Solid lines: output variance gain $(N^2)$ and bias (expected value) of output $(M)$ predicted by Jacobian and Stochastic linearizations with different levels of input variance for (**left**) **tanh**, (**middle**) **softplus**, (**right**) **ReLU**. Dashed lines: true variance evaluated empirically by generating $10^5$ noise samples at each zero-noise input. The higher the input variance, the more significant is the benefit of using SL. Note there is a difference in input variance sensitivity for the three transformations: SL has no significant advantage over JL when input variance has an order of magnitude of $\leq -2, \leq -1, \leq -3$ for tanh, softplus, and ReLU (dark grey lines).

We then construct a four layer feed-forward model to compare the theoretically predicted (posterior) distribution to the true (empirical) one. Each layer has 3, 100, 100, and 2 units, respectively. We randomly initialize the weights, bias, and latent variable $x$ from uniform distributions, and input noise from multivariate Gaussian distributions with 0.16 variance and random correlation. Figure 3 shows three examples in which our SL model makes compelling prediction of both location and variance-covariance matrix of the output, even though the output do not follow Gaussian distributions. By contrast, JL model produces significant inaccuracy on both metrics. We purposefully choose low dimensional input and output space in this experiment for accurate comparison. Though SL is not limited to low-D cases, obtaining the true output covariance, which requires a non-sparse sampling of the input space, is subject to the power law. For example, it requests at least $500^{784}$ sample points out of a 784-variate Gaussian distribution to estimate the true output covariance of a simple neural network trained on MNIST test, and down-sampling input will lead to a sparse visiting of the output space.

## 3.2 TIME-ACCURACY TRADE-OFF BETWEEN JL, SL, AND SAMPLING METHOD

We compared the calculation time and predictive accuracy of covariance, defined by Correlation Matrix Distance (CMD) Herdin et al. (2005), and of mean, defined as Euclidean distance from true mean, between SL and Monte Carlo Sampling (MC).

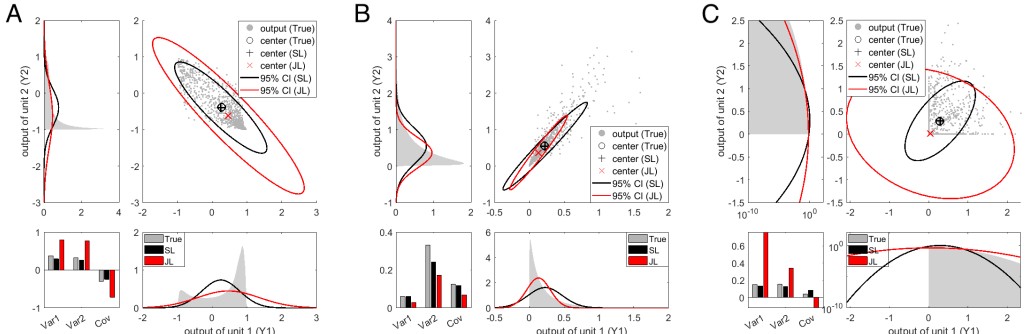

Figure 3: Comparing theoretical prediction to empirical outputs. **(A) tanh (B) softplus (C) ReLU**. For each panel: **Top right**: Grey scatters are empirical output given by the network. Though only 1000 scatters are displayed, the center is obtained by sampling $10^8$ realizations. The positions of centers reveal promising accuracy of SL model on output mean. Black and red ellipses correspond to the 95% credible intervals of SL's and JL's predictive distributions. **Bottom right and top left**: Empirical pdf (histogram) of MLP output v.s. theoretical prediction (solid lines) by SL and JL models. **Bottom left**: Comparing empirical variance-covariances and predicted ones by the two methods.

Table 1: Time-accuracy trade-off between linearization and sampling (MC) methods

|  | JL | SL | MC100 | MC500 | MC2500 | MC12500 |
| --- | --- | --- | --- | --- | --- | --- |
| Time (100 repeats) (s) | 0.0795 | **0.1181** | 0.1817 | **0.4597** | 1.7809 | 8.6177 |
| CMD | 0.1011 | **0.0046** | 0.0640 | **0.0066** | 7.8511e-4 | 1.2547e-4 |
| Eucl. Dis. | 5.4443 | **0.4440** | 1.1151 | **0.4844** | 0.1753 | 0.0948 |

To address the concern of scalability, we train a 784 x 256 x 256 x 256 x10 network on MNIST dataset LeCun & Cortes (2010), with **ReLU**, **softplus**, and **tanh** intermediate layers, and test performance on estimating propagated uncertainty. To minimize the influence on computing time caused by different software implementations and optimization (of MC and SL), we limit the computing resource to one CPU core. The result shows to have the same predictive accuracy of covariance as SL, MC needs more than 500 samples, about five times slower than SL (Table 1). It worth noting that the accuracy of sampling methods also owes to the sparse sampling (1e7 samples out of 784 dim. space). We believe the real value of output covariance is closer to SL's prediction.

### 3.3 Using SL to enable Kalman filtering for target tracking

In the previous sections, we established that quaslinear approximation can accurately estimate uncertainty propagation in neural networks. In this section, we test whether calculating this uncertainty, combined with Bayesian/Kalman Filtering Kálmán (1960) can further improve performance in integrated systems, as motivated in the Introduction.

We test these benefits in the context of target-tracking with an aerial mobile camera (e.g., a drone; Fig. 4A). The task is to track the ground-level target's geographic position using noisy images captured by the camera. In real-life applications, the camera's exact position and velocity are unknown due to the limited resolution of GPS and environmental forces such as wind. Likewise, images are obscured by environmental noise and distorted according to the camera's velocity. Thus, uncertainty in target position arises due to both the camera's position (reference frame) and various image artifacts (Fig. 4B).

We model these processes using autoregressive random-walks for the target and drone kinematics with additional random perturbations:

$$x_{targ}(t+1) = \beta_{targ}x_{targ}(t) + v_{targ}(t) + \omega_{targ}(t) \tag{16}$$

$$v_{targ}(t+1) = \theta_{targ}v_{targ}(t) + \epsilon_{targ}(t) \tag{17}$$

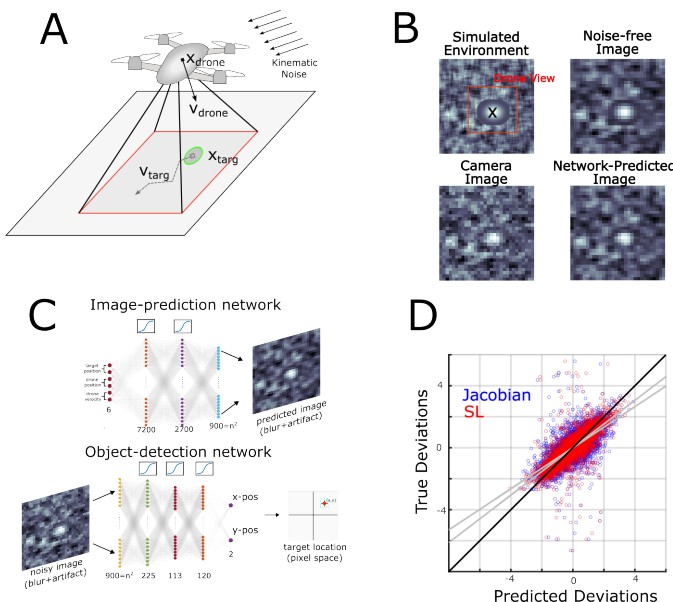

Figure 4: Schematic and results of simulated aerial object-tracking. A) Problem-setup: a mobile aerial camera attempts to track the geographic position of a moving target. B) Pre-noise images reflect cropping of the true environment to the camera view, downsampling to the camera-resolution and motion blur (upper-right). Additional autoregressive environmental noise further contaminates images (bottom left). The image-prediction network is used to generate a distribution of noise-free camera images (bottom right) as a function of environmental states (positions/velocities). C) Architectures of the image-prediction ($6 \times 7200 \times 2700 \times 900$) (top) and object-detection (bottom) networks ($900 \times 225 \times 113 \times 120 \times 2$). D) Example of true and predicted deviations in target position (from mean) due to Jacobian and Stochastic Linearization end-to-end (from the environmental distributions to image generation to target positions in pixel-space). Units are z-scored.

and similarly for the drone kinematics (using $\beta_{drone}, x_{drone}$, etc.) The state vectors $x(t), v(t) \in \mathbb{R}^2$ reflect position and velocity in geographic (2D) coordinates. Similarly, the process perturbations ($\omega, \epsilon \in \mathbb{R}^2$) reflect perturbations in this plane, whereas the constants $\beta, \theta$ are scalars. Camera images were simulated by first placing a target, according to its geographic position on a static background. The image was then cropped to a square centered at the drone position blurred with a motion kernel and downsampled by a factor of 3. This distorted image is denoted $\hat{z}_t \in \mathbb{R}^{n \times n}$ for an image with $n \times n$ pixels. Camera/environmental noise was modeled as the sum of additive spatially-autoregressive Gaussian process:

$$z(t) = \hat{z}(t) + M\eta(t) + \hat{\eta}(t) \tag{18}$$

with $z, \hat{z}\hat{\eta} \in \mathbb{R}^{n \times n}$, denoting the observed-image, pre-noise image, and speckle-style (spatially-independent) noise, respectively. The composition $M\eta(t)$ denotes the Gaussian blurring ($M$) of a $n \times n$ matrix with independent, random entries ($\eta(t)$).

The target consisted of an elliptical sinc function oriented along the x axis, cropped to an ellipse. Additional details regarding the target generation and environment-generation are provided in SI.

We estimated the distribution of possible camera-images due to environmental states (drone/target position and velocity) by training a network to predict (noise-free) camera images as a function of environmental states (Fig. 4C, top). This image-prediction network was feedforward, fully-connected with tanh-activation and regression (linear) input and output.

The target's 2D location, relative the drone, was estimated by a fully-connected regression network with **tanh**-activation and layer-sizes as indicated in Fig. 4C, bottom. The network was trained to predict the position of the target within each image (i.e., relative to the target: $x_{targ} - x_{drone}$) using 25,000 simulated environments with 10 simulated camera images ($z_t$) each (250,000 total examples). The target was always fully contained within the camera images. Additional training details are provided in SI.

We denote the network $\zeta : \mathbb{R}^{n \times n} \to \mathbb{R}^2$, hence the relative position of the target is predicted to be $\zeta(z_t)$. In addition to the target, noisy estimates are also available for the drone's position and velocity. We combine these measurements into the sensor-output vector ($y_t \in \mathbb{R}^6$):

$$y_t = \begin{bmatrix} \zeta(z(t)) \\ x_{drone}(t) + \epsilon_x(t) \\ v_{drone}(t) + \epsilon_v(t) \end{bmatrix} \tag{19}$$

The Bayesian Filter (or Kalman Filter) is an iterative algorithm that seeks to estimate the hidden states of a dynamical system. Given a stochastic dynamical systems model:

$$x(t+1) = f(x(t), \omega(t)) \tag{20}$$

$$y(t+1) = h(x(t+1), \epsilon(t)) \tag{21}$$

The Kalman Filter estimates $x$ from measurements $y$ according to the algorithm:

$$\hat{x}_{t+1|t} = \mathbb{E}[f(x)|\hat{x}_t, \hat{P}_t] \tag{22}$$

$$\hat{P}_{t+1|t} = \mathbf{Cov}[f(x)|\hat{x}_t, \hat{P}_t] \tag{23}$$

$$K = \mathbf{Cov}[x_{t+1}, y_{t+1}]\mathbf{Cov}[y_{t+1}]^{-1} \tag{24}$$

$$\hat{x}_{t+1} = \hat{x}_{t+1|t} + K(\mathbb{E}[y_{t+1}|\hat{x}_{t+1|t}, \hat{P}_{t+1|t}] - y_{t+1}) \tag{25}$$

$$\hat{P}_{t+1} = \hat{P}_{t+1|t} - K\mathbf{Cov}[y_{t+1}]K^{\top}. \tag{26}$$

Here, $\hat{x}$ and $\hat{P}$ denote the expectation and covariance for the unknown system-state $x$. The subscript $t+1|t$ denotes the Bayesian prior estimate, whereas $t+1$ denotes the Bayesian-posteriors for $x_{t+1}$. For a linear system:

$$x_{t+1} = A_t x_t + \omega_t \tag{27}$$

$$y_{t+1} = H_{t+1} x_{t+1} + \epsilon_{t+1} \tag{28}$$

$$\hat{P}_{t+1|t} = A_t \hat{P}_t A_t^{\top} + \mathbf{Cov}[\omega_t] \tag{29}$$

$$\mathbf{Cov}[y_t] = H_{t+1} \hat{P}_{t+1|t} H_{t+1}^{\top} + \mathbf{Cov}[\epsilon_{t+1}] \tag{30}$$

$$\mathbf{Cov}[x_{t+1}, y_{t+1}] = \hat{P}_{t+1|t} H_{t+1}^{\top} \tag{31}$$

Similarly, the quasi-linear form for nonlinear measurements replaces the measurement function $h$ with its stochastic linearization:

$$H_{t+1} = \mathbb{E}\left[\frac{\partial h(x_{t+1}, \epsilon_{t+1})}{\partial x_{t+1}}\right] \tag{32}$$

$$M_{t+1} = \mathbb{E}\left[\frac{\partial h(x_{t+1}, \epsilon_{t+1})}{\partial \epsilon_{t+1}}\right] \tag{33}$$

$$\mathbf{Cov}[y_{t+1}] = H_{t+1} \hat{P}_{x+1|t} H_{t+1}^{\top} + M_{t+1}\mathbf{Cov}[\epsilon_{t+1}]M_{t+1}^{\top} \tag{34}$$

but retains the full nonlinearity for $\mathbb{E}[y_{t+1}]$.

We tested two aspects of the quasilinearization for these networks: (1) the statistical accuracy of the quasilinear approximation and (2) improvements in target-tracking performance by integrating these networks with Bayesian state-estimation (Kalman filtering).

In the first test, we estimated how well the quasilinear approximation, as opposed to Jacobian methods, estimated the distribution of predicted images and sensor predictions as a function of environmental parameters (drone position/velocity and target position). For these tests, we used 51 distributions of the drone/target environment with 2,500 sampled camera-images each (details in SI) to compare true and estimated statistics. Results demonstrated consistently better mapping between environmental distributions and predictions of the object-detection network in pixel-space (example Fig. 4D). The quasi-linearization improved this correlation by $.017 \pm .022 (n = 51)$, $paired - t(50) = 5.52, p \approx E - 6$ and improved prediction of the expected image-location in pixel space: difference in Euclidean-norm $= .74 \pm .75, p \approx 6E - 9$.

We then integrated neural network SL into an end-to-end Kalman-filter and compared these results to a Kalman-filter which did-not model uncertainty-propagation through the object-detection

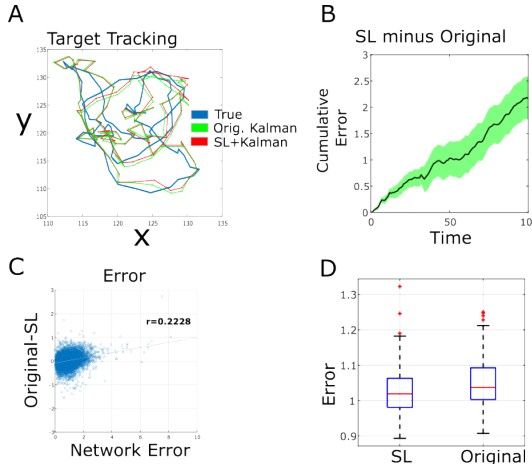

Figure 5: Improved target-tracking due to stochastic linearization (SL) of network uncertainty. A) Representative (closest to mean loss) trajectory of true and predicted target positions in geographic space. B) Time-course of cumulative difference in error (root-mean-square) between Kalman filtering with/without stochastic (SL). C) Scatterplot of error in estimating target position with vs. without SL. D) Boxplot of total-error for each simulation (averaging over time)

network (i.e., treated the estimated object locations as veridical). Results demonstrate improved accuracy of target-tracking when including variance generated by the neural network (Fig. 5A). Improvements reflected a sustained performance boost as opposed to transient resolution of initial uncertainty (Fig. 5B). As expected, we found that improvements were greatest during periods when the object-detection network was least accurate in locating the target within camera images (Fig. 5C; $r(9998) = .22$). These factors led to improved overall performance when propagating uncertainty through the neural network: MSE difference=$.027 \pm .04, p < 5.5 \times 10^{-7}$. Together these results suggest that stochastic linearization is a potent tool for estimating how environmental uncertainty propagates through nonlinear (neural) networks and consequently can be leveraged to improve state-estimates.

## 4 CONCLUSIONS

In this article, we develop a quasilinear approximation approach based on stochastic linearization to estimate uncertainty propagating through neural networks. The proposed method lends significant advantages in terms of enabling analytic estimation techniques, such as Kalman filtering, for estimating latent dynamics upstream of sensors. One envisioned use-case for this methdology is in the target tracking domain. We derived the analytic solutions for the equivalent gains and bias for the SL realization of a MLP with **tanh** activations, and illustrate its accuracy in predicting mean and covariance. We then verify the effectiveness of SL-Kalman filter coupling with a detailed simulation example, where a drone tracks a point target in a dynamic environment. However, some limitations are worth noting. Despite the strength of SL, when the true output distribution manifests higher order moments due to extensive nonlinear effects, the SL prediction is limited since the Gaussian approximation will become increasingly inaccurate in this setting. Indeed, here we focus on the first and second order moment of predictive distributions (mean and variance), and consideration of higher orders of moment (skewness, kurtosis, etc.) is beyond the predictive power of linear approximation. Additionally, SL tends to underestimate covariance when input variance is large ; on the other hand, if the input variance shrinks to small values, using SL becomes no more beneficial than JL. This could happend near the diminishing tails of activation functions. Despite these limitations, the proposed method would provide significant gains in real-time deployments, where large-scale sampling approaches would be computationally intractable.

## 5 CODE AVAILABILITY

The code of generating the figures will be publicly available after the reviewing session.

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
