## A APPENDIX

### A.1 GENERAL SYMBOLS

Revisit the meaning of symbols.

| | | |
|---:|---|---|
| $x$ | input latent variables | |
| $\xi$ | multivariate Gaussian noises (injected to $x$) | $\sim \mathcal{N}^n(\mathbf{0}, \Sigma)$ |
| $w$ | input weight vector | |
| $b$ | bias vector | |
| $u$ | input of an activation function, equals $w^\top(x + \xi)$ | |
| $\tilde{\sigma}^2$ | variance of $u$, equals $w^\top \Sigma w$ | |
| $\{\gamma_1, ..., \gamma_p\}$ | scaling factor set. e.g. $\{0.64, 1.13\}$ | |
| subscript $i$ | $i$-th unit of a layer | |
| superscript $(k)$ | $k$-th layer of a network | |

For simplicity of the derivation, denote:

$$z \triangleq w^\top x + b$$
$$\tau \triangleq w^\top \xi$$

where $z$ represents the zero-noise component (center) inside the activation function, and $\tau$, the uncertainty component. Then,

$$u + b = z + \tau$$

The following results depend on the fact that Gaussian distribution is *stable* – a linear combination of Gaussian random variables $\sim \mathcal{N}^n(0, \Sigma)$ is still a Gaussian random variable $\sim \mathcal{N}(0, w^\top \Sigma w)$.

### A.2 DERIVATION OF QUASI-LINEAR GAIN AND BIAS FOR **tanh** (AND SIGMOIDAL) LAYERS

We use a linear combination of error functions with different scaling factors to approximate tanh function. In our experiments, a set of two scaling parameters, $\{0.64, 1.13\}$, is enough to maintain a favorable error. In practice, one can add more terms for even higher accuracy without losing efficiency (depending on the computing resources), because the extra terms can be easily paralleled.

$$\mathbf{tanh}(u + b) \approx \frac{1}{p} \sum_{j=1}^{p} \mathbf{erf}\left[\gamma_j(u + b))\right]$$

Thus, the quasilinear gain $N$ and bias $M$ of a single unit in a **tanh** layer are

$$N = \mathbb{E}\left[\frac{\mathrm{d}}{\mathrm{d}u}\left(\frac{1}{p}\sum_{j=1}^{p}\mathbf{erf}\left[\gamma_j(u + b))\right]\right)\right]$$

$$M = \mathbb{E}\left[\frac{1}{p}\sum_{j=1}^{p}\mathbf{erf}\left[\gamma_j(u + b))\right]\right] \tag{35}$$

To find quasilinear gain $N$

$$\mathbb{E}[\frac{\mathrm{d}}{\mathrm{d}u}\mathbf{erf}(\gamma(u + b))]$$

$$= \int \left[\frac{\mathrm{d}}{\mathrm{d}u}\mathbf{erf}(\gamma(u + b))\right]p(\tau)\mathrm{d}\tau$$

$$= \int_{-\infty}^{\infty} \frac{2\gamma}{\sqrt{\pi}}\mathbf{exp}\left[-\gamma^2(z + \tau)^2\right]\frac{1}{\sqrt{2\pi}\tilde{\sigma}}\mathbf{exp}\left[-\frac{\tau^2}{2\tilde{\sigma}^2}\right]\mathrm{d}\tau$$

$$= \frac{\sqrt{2}\gamma}{\pi\tilde{\sigma}}\int_{-\infty}^{\infty}\mathbf{exp}\left[-\gamma^2(z + \tau)^2 - \frac{\tau^2}{2\tilde{\sigma}^2}\right]\mathrm{d}\tau$$

complete the square

$$= \frac{\sqrt{2}\gamma}{\pi\tilde{\sigma}} \int_{-\infty}^{\infty} \exp\left[-\left(\sqrt{\frac{1}{2\tilde{\sigma}^2}+\gamma^2}\,\tau + \frac{\gamma^2 z}{\sqrt{\frac{1}{2\tilde{\sigma}^2}+\gamma^2}}\right)^2 + \frac{\gamma^4 z^2}{\frac{1}{2\tilde{\sigma}^2}+\gamma^2} - \gamma^2 z^2\right]\mathrm{d}\tau$$

use substitution $\quad \nu = \sqrt{\frac{1}{2\tilde{\sigma}^2}+\gamma^2}\,\tau + \frac{\gamma^2 z}{\sqrt{\frac{1}{2\tilde{\sigma}^2}+\gamma^2}} \quad$ then $\quad \mathrm{d}\nu = \sqrt{\frac{1}{2\tilde{\sigma}^2}+\gamma^2}\,\mathrm{d}\tau$

$$= \frac{\sqrt{2}\gamma}{\pi\tilde{\sigma}} \int_{-\infty}^{\infty} \frac{1}{\sqrt{\frac{1}{2\tilde{\sigma}^2}+\gamma^2}} \exp\left[-\nu^2 + \frac{\gamma^4 z^2}{\frac{1}{2\tilde{\sigma}^2}+\gamma^2} - \gamma^2 z^2\right]\mathrm{d}\nu$$

$$= \frac{\gamma}{\sqrt{2\pi}\tilde{\sigma}\sqrt{\frac{1}{2\tilde{\sigma}^2}+\gamma^2}} \exp\left[\frac{\gamma^4 z^2}{\frac{1}{2\tilde{\sigma}^2}+\gamma^2} - \gamma^2 z^2\right] \int_{-\infty}^{\infty} \frac{2}{\sqrt{\pi}} \exp(-\nu^2)\mathrm{d}\nu$$

since $\int_{-\infty}^{\infty} \frac{2}{\sqrt{\pi}}\exp(-\nu^2)\mathrm{d}\nu = 2$ , simplify and get

$$= \frac{2}{\sqrt{\pi}} \frac{\gamma}{\sqrt{2\gamma^2\tilde{\sigma}^2+1}} \exp\left[-\frac{\gamma^2 z^2}{2\gamma^2\tilde{\sigma}^2+1}\right]$$

$$= \frac{2}{\sqrt{2\pi}} \left(\frac{2\gamma^2\tilde{\sigma}^2+1}{2\gamma^2}\right)^{-\frac{1}{2}} \exp\left[-\frac{1}{2}\frac{z^2}{(2\gamma^2\tilde{\sigma}^2+1)/(2\gamma^2)}\right]$$

which can be rewritten with standard Gaussian pdf $\varphi(\cdot)$

$$= \frac{2}{\hat{\sigma}}\,\varphi\left(\frac{z}{\hat{\sigma}}\right)$$

where $\hat{\sigma}^2 = \frac{2\gamma^2\tilde{\sigma}^2+1}{2\gamma^2}$. Therefore,

$$N = \frac{1}{p}\sum_{j=1}^{p} \mathbb{E}\left[\frac{\mathrm{d}}{\mathrm{d}u}\mathbf{erf}\left[\gamma_j(u+b)\right]\right] = \frac{2}{p}\sum_{j=1}^{p}\frac{1}{\hat{\sigma}_j}\,\varphi\left(\frac{z}{\hat{\sigma}_j}\right)$$

where

$$\hat{\sigma}_j^2 = \frac{2\gamma_j^2\tilde{\sigma}^2+1}{2\gamma_j^2}$$

To find the quasilinear bias $M$,

$$\mathbb{E}[\mathbf{erf}(\gamma(u+b))]$$

$$= \int \mathbf{erf}(\gamma(u+b))p(\tau)\mathrm{d}\tau$$

$$= \int \frac{\mathrm{d}}{\mathrm{d}u}\left[\int \mathbf{erf}(\gamma(u+b))p(\tau)\mathrm{d}\tau\right]\mathrm{d}u$$

$$= \int\int\left[\frac{\mathrm{d}}{\mathrm{d}u}\mathbf{erf}(\gamma(u+b))\right]p(\tau)\mathrm{d}\tau\mathrm{d}u$$

$$= \int \mathbb{E}\left[\frac{\mathrm{d}}{\mathrm{d}u}\mathbf{erf}(\gamma(u+b))\right]\mathrm{d}u$$

since $u = z + \tau - b$, $\mathrm{d}u = \mathrm{d}z$

also $\frac{d}{du}\mathbf{erf}(\gamma(u+b)) = \frac{d}{dz}\mathbf{erf}(\gamma(u+b))$ , then

$$= \int_0^z \frac{2}{\hat{\sigma}} \, \boldsymbol{\varphi}\left(\frac{z}{\hat{\sigma}}\right) \, dz$$

$$= 2\boldsymbol{\Phi}\left(\frac{z}{\hat{\sigma}}\right) - 1 \quad \text{where } \boldsymbol{\Phi}(\cdot) \text{ is the standard Gaussian cdf}$$

$$= \mathbf{erf}\left(\frac{z}{\sqrt{2\hat{\sigma}^2}}\right)$$

Therefore,

$$M = \frac{1}{p}\sum_{j=1}^p \mathbb{E}\left[\mathbf{erf}(\gamma_j(u+b))\right] = \frac{1}{p}\sum_{j=1}^p \mathbf{erf}\left(\frac{z}{\sqrt{2\hat{\sigma}_j^2}}\right)$$

The derivation of SL model for the whole $\mathbf{tanh}$ layers is similar to its single unit counterpart (see below). However, one should note when replacing weight vector $w$ with weight matrix (of $k^{th}$ layer) $w^{(k)}$, the $\mathbf{exp}$ terms yield matrices, but we only need to calculate the diagonal entries. Lastly, the SL model of layers with sigmoidal activations (e.g. logistic) can be obtained in a similar way.

$$N^{(k)} = \frac{2}{p}\sum_{j=1}^p \frac{1}{\hat{\sigma}_j^{(k)}} \odot \boldsymbol{\varphi}\left(\frac{z^{(k)}}{\hat{\sigma}_j^{(k)}}\right) \tag{36}$$

$$M^{(k)} = \frac{1}{p}\sum_{j=1}^p \mathbf{erf}\left(\frac{z^{(k)}}{\sqrt{2\hat{\sigma}_j^{(k)^2}}}\right) \tag{37}$$

where

$$\hat{\sigma}_j^{(k)^2} = \frac{2\gamma_j^2\tilde{\sigma}^{(k)^2} + 1}{2\gamma_j^2} \quad \text{and} \quad \tilde{\sigma}^{(k)^2} = \mathbf{diag}\left(w^{(k)\top}\Sigma^{(k-1)}w^{(k)}\right) \tag{38}$$

### A.3 DERIVATION OF QUASI-LINEAR GAIN AND BIAS FOR $\mathbf{softplus}$ LAYERS

The derivation of quasi-linear gain and bias for a $\mathbf{softplus}$ layer is related to that for a $\mathbf{sigmoid}$ layer (section 5.1), since the derivative of the $\mathbf{softplus}$ function is the $\mathbf{sigmoid}$ function, and the latter can be approximated with a linear combination of Gaussian cdf. We have

$$\mathbf{softplus}(u+b) = \frac{1}{\beta}\mathbf{log}(1 + e^{\beta(u+b)})$$

For simplicity, we choose $\beta = 1$ as default value. Then we use the following approximation

$$\frac{d}{du}\mathbf{softplus}(u+b) = \frac{1}{1 + e^{-(u+b)}} \approx \frac{1}{p}\sum_{j=1}^p \boldsymbol{\Phi}\left(\frac{\gamma_j}{\sqrt{2}}(u+b)\right)$$

Here $\boldsymbol{\Phi}(\cdot)$ is the standard normal cdf, which can be rewritten by error function

$$\boldsymbol{\Phi}\left(\frac{\gamma}{\sqrt{2}}(u+b)\right) = \frac{1}{2}\left(1 + \mathbf{erf}\left(\frac{\gamma(u+b)}{2}\right)\right)$$

There are two reasons that we use the error function. The first is to reuse the derivation of $\mathbb{E}[\mathbf{erf}(\gamma(u+b))]$ in the previous section; the second is that calculating Gaussian cdf is computationally demanding, while the approximation algorithm of the error function is available [Cody (1969)]. We also convert our final answer to complementary error function $\mathbf{erfc}$ to avoid subtractive cancellation that leads to inaccuracy in the tails. Now the quasilinear gain $N$ becomes

$$N = \mathbb{E}\left[\frac{d}{du}\mathbf{softplus}(u+b)\right] \approx \frac{1}{p}\sum_{j=1}^p \boldsymbol{\Phi}\left(\frac{z}{\hat{\sigma}_j}\right) = \frac{1}{2p}\sum_{j=1}^p \mathbf{erfc}\left(-\frac{z}{\sqrt{2\hat{\sigma}_j^2}}\right)$$

where

$$\hat{\sigma}_j^2 = \frac{2\left(\frac{\gamma_j}{2}\right)^2 \tilde{\sigma}^2 + 1}{2\left(\frac{\gamma_j}{2}\right)^2} = \frac{\gamma_j^2 \tilde{\sigma}^2 + 2}{\gamma_j^2}$$

Therefore,

$$
\begin{aligned}
M &= \mathbb{E}\left[\mathbf{softplus}(u + b)\right] \\
&= \int \mathbf{softplus}(u + b)\, p(\tau)\, \mathrm{d}\tau \\
&= \int \frac{\mathrm{d}}{\mathrm{d}u}\left[\int \mathbf{softplus}(u + b) p(\tau)\mathrm{d}\tau\right] \mathrm{d}u \\
&= \int \int \left[\frac{\mathrm{d}}{\mathrm{d}u}\mathbf{softplus}(u + b)\right] p(\tau)\mathrm{d}\tau \mathrm{d}u \\
&= \int \mathbb{E}\left[\frac{\mathrm{d}}{\mathrm{d}u}\mathbf{softplus}(u + b)\right] \mathrm{d}u \\
&= \int_{-\infty}^{z} \frac{1}{p} \sum_{j=1}^{p} \mathbf{\Phi}\left(\frac{z}{\hat{\sigma}_j}\right) \mathrm{d}z \\
&= \frac{1}{p} \sum_{j=1}^{p} \int_{-\infty}^{z} \mathbf{\Phi}\left(\frac{z}{\hat{\sigma}_j}\right) \mathrm{d}z \\
&= \frac{1}{p} \sum_{j=1}^{p} \left[z\, \mathbf{\Phi}\left(\frac{z}{\hat{\sigma}_j}\right) + \hat{\sigma}_j \varphi\left(\frac{z}{\hat{\sigma}_j}\right)\right] \\
&= \frac{1}{p} \sum_{j=1}^{p} \left[\frac{z}{2}\, \mathbf{erfc}\left(-\frac{z}{\sqrt{2\hat{\sigma}_j^2}}\right) + \hat{\sigma}_j \varphi\left(\frac{z}{\hat{\sigma}_j}\right)\right] \\
&= Nz + \frac{1}{p} \sum_{j=1}^{p} \hat{\sigma}_j \varphi\left(\frac{z}{\hat{\sigma}_j}\right)
\end{aligned}
$$

is the quasilinear bias, where $\varphi(\cdot)$ is the standard Gaussian pdf. Rewrite for the full layer and get

$$N^{(k)} = \frac{1}{2p} \sum_{j=1}^{p} \mathbf{erfc}\left(-\frac{z^{(k)}}{\sqrt{2\hat{\sigma}_j^{(k)2}}}\right) \tag{39}$$

$$M^{(k)} = N^{(k)} \odot z^{(k)} + \frac{1}{p} \sum_{j=1}^{p} \hat{\sigma}_j^{(k)} \odot \varphi\left(\frac{z^{(k)}}{\hat{\sigma}_j^{(k)}}\right) \tag{40}$$

where

$$\hat{\sigma}_j^{(k)2} = \frac{\gamma_j^2 \tilde{\sigma}^{(k)2} + 2}{\gamma_j^2} \quad \text{and} \quad \tilde{\sigma}^{(k)2} = \mathbf{diag}\left(w^{(k)\top} \Sigma^{(k-1)} w^{(k)}\right) \tag{41}$$

## A.4 DERIVATION OF QUASI-LINEAR GAIN AND BIAS FOR RELU LAYERS

First, consider a single $i$-th unit in a ReLU layer. Let $z_i = w_i^\top x + b_i$, then the gain of the $i^{th}$ unit, $N_i$, is

$$
\begin{aligned}
N_i &= \mathbb{E}\left[\frac{\mathrm{d}}{\mathrm{d}u}\mathbf{ReLU}(u)\right] \\
&= \int_{-z_i}^{\infty} p(\tau_i)\mathrm{d}\tau_i \\
&= 1 - \mathbf{\Phi}(-\frac{z_i}{\tilde{\sigma}_i}) \\
&= \frac{1}{2}\left[1 + \mathbf{erf}(\frac{z_i}{\sqrt{2\tilde{\sigma}_i^2}})\right]
\end{aligned}
$$

where $\tilde{\sigma}_i^2 = w_i^\top \Sigma w_i$.

The bias of the $i^{th}$ unit, $M_i$, is

$$
\begin{aligned}
M_i &= \mathbb{E}[\mathbf{ReLU}(u)] \\
&= \int_{-z_i}^{\infty} (z_i + \tau_i)\, p(\tau_i)\mathrm{d}\tau_i \\
&= \underbrace{(z_i) \int_{-z_i}^{\infty} p(\tau_i)\mathrm{d}\tau_i}_{A} + \underbrace{\int_{-z_i}^{\infty} \tau_i\, p(\tau_i)\mathrm{d}\tau_i}_{B}
\end{aligned}
$$

$A = N_i z_i$

$B = $ mean of the unnormalized truncated Gaussian $\mathcal{N}(\tau \mid 0, w_i^\top \Sigma w_i, -z_i, \infty)$

$$
= \tilde{\sigma}_i\, \varphi\left(-\frac{z_i}{\tilde{\sigma}_i}\right)
$$

Rewrite for the full layer and get

$$
N^{(k)} = \frac{1}{2}\left[1 + \mathbf{erf}\left(\frac{z^{(k)}}{\sqrt{2\tilde{\sigma}^{(k)2}}}\right)\right] \tag{42}
$$

$$
M^{(k)} = N^{(k)} \odot z^{(k)} + \tilde{\sigma}^{(k)} \odot \varphi\left(-\frac{z^{(k)}}{\tilde{\sigma}^{(k)}}\right) \tag{43}
$$

where $z^{(k)} = w^{(k)\top} M^{(k-1)} + b^{(k)}$, and

$$
\tilde{\sigma}^{(k)2} = \mathbf{diag}(w^{(k)\top} \Sigma^{(k-1)} w^{(k)}) \tag{44}
$$

The SL model of layers with any piece-wise linear activation function (e.g. leaky ReLU) can be obtained in a similar way.

## A.5 KALMAN SIMULATION DETAILS

The simulated 2D drone environment consisted of a $180\times180$ grid generated according to:

$$
B_{m\times m} = 1_{2\times 2} \otimes B_1 + G \otimes B_2^3 \tag{45}
$$

in which $B_1$ and $B_2$ are matrices with normally distributed independent entries, $1_{2\times 2}$ denotes the $2\times 2$ box-smoothing filter and $G$ denotes the Gaussian filter with standard-deviation 2-pixels. The background ($B$) was then re-scaled to have a maximum absolute value of 3.

The target consisted of a cropped elliptical sinc-function:

$$
T(x,y) = 1.5\mathbf{sinc}\left(1.5\sqrt{\left(\frac{x}{16}\right)^2 + \left(\frac{y}{14}\right)^2}\right) \tag{46}
$$

with integers $x \in [-16, 16]$ and $y \in [-14, 14]$. The function was then cropped to a (discretized) ellipse according to the selection rule:

$$\left(\frac{x}{16}\right)^2 + \left(\frac{y}{14}\right)^2 \le 1.25 \tag{47}$$

We incorporated targets into the environment by rounding the target's continuous-valued position and placing the cropped-target image at that location in the environment (on-center).

Both the image generation and object-detection networks were trained using 250,000 exemplars featuring the same target used in testing. The image-generation network was trained to reproduce images in the same environment as testing, whereas the object-detection network was trained using 25,000 simulated environments with 10 images each to reflect a generic sensing network.

We used the image-generation network for static (single time-point) testing in the main text as the combined image-generation and object-detection networks contain fan-out and fan-in architectures, respectively. This test improves generality to other network architectures. However, in application, it is more accurate and tractable to directly model the relationship between true states and network outputs with a single network (i.e., how position and kinematics bias the detected target location), as opposed to first simulating the predicted image (the image-generation network) and then passing it to the object-detection network. Therefore, for testing a dynamic target, we trained a simple network ("auxiliary network") to directly predict the object-detection network's output using the same inputs (drone location, target location, and drone velocity). This network contained two fully-connected tanh-layers (40 and 60 units) with fully-connected regression layers for input/output.

All networks were trained in MATLAB R2022B's Deep Learning package to minimize $L_2$ loss using ADAM with default parameters ($\beta_1 = .9$, $\beta_2 = .99$) and rate $= 2 \times 10^{-4}$. Minibatches contained 1000 images and 5,000 additional images were held-out for cross-validation, hence each training epoch contained 250-folds (minibatches). The number of training epochs was tuned based upon visual inspection of cross-validation loss: 100 epochs (25,000 minibatches) for image-generation, 350 epochs (87,500 minibatches) for object-detection, and 500 epochs (125,000 minibatches) for the auxiliary network.

In the static (1-step) simulations in the main text kinematics were ignored as the 1-step prior-distributions do not interact with method (i.e. $x_{SL}(0) = x_{Jac}(0)$) implies ($Ax_{SL}(0) = Ax_{Jac}(0)$). Instead distributions were randomly generated according to:

$$\sqrt{P_{1|0}} = (M_0 + \frac{I_8}{2})\sqrt{\Sigma} \tag{48}$$

with $M$ randomly generated for each exemplar (normal-iid) and scaling factor $\sqrt{\Sigma}=.4$ for target/drone velocity and 1 for target/drone position. $\sqrt{P}$ denotes an arbitrary matrix root ($P = \sqrt{P}\sqrt{P}^T$).

In the dynamic simulation, we modified kinematics to be mean-reverting (ensuring that the drone/target did not fly out of bounds). The full kinematics were:

$$z := \begin{bmatrix} z_{drone} \\ z_{targ} \end{bmatrix} := \begin{bmatrix} pos_{drone-x} \\ pos_{drone-y} \\ vel_{drone-x} \\ vel_{drone-y} \\ pos_{targ-x} \\ pos_{targ-y} \\ vel_{targ-x} \\ vel_{targ-y} \end{bmatrix} \tag{49}$$

$$z_{drone}(t+1) = \begin{bmatrix} (1-\alpha_D)I_2 & .25I_2 \\ -\beta_D I_2 & .9I_2 \end{bmatrix} + \frac{m-k}{2} \begin{bmatrix} \alpha_D \\ \alpha_D \\ \beta_D \\ \beta_D \end{bmatrix} + \eta_{drone} \tag{50}$$

$$z_{targ}(t+1) = \begin{bmatrix} (1-\alpha_T)I_2 & .25I_2 \\ -\beta_T I_2 & .9I_2 \end{bmatrix} + \frac{m}{2} + \begin{bmatrix} \alpha_T \\ \alpha_T \\ \beta_T \\ \beta_T \end{bmatrix} + \eta_{targ} \tag{51}$$

with $\alpha_D = .1, \beta_D = .01, \alpha_T = .1, \beta_T = .05$, $m = 240$ denoting the (square) environment's width and $k$ denoting the square field-of-view width for the drone's camera. $\eta_{drone}$ was Gaussian distributed with var = 2 for position and .6 for velocity. $\eta_{targ}$ was the sum of a Gaussian process (var=.4 for position and 1 for velocity) and the 5-step moving-average of a Gaussian process following the same distribution.

In addition to camera-images, the drone received simulated onboard sensor-estimates (i.e., GPS) of its current position and velocity with noise variances of 1 and .5, respectively. These readings were combined with the image measurements during Kalman filtering. Motion blur was simulated using the MATLAB image-processing toolbox using polar-coordinates of drone velocity to parameterize the angle and magnitude of the motion kernel.