# OpenReview forum: "Estimating uncertainty from feed-forward network based sensing using quasilinear approximation"
_ICLR.cc/2024/Conference — Submitted to ICLR 2024_

### Official Review · Reviewer_vnNG · 2023-10-31

**Soundness:** 2 fair
**Presentation:** 2 fair
**Contribution:** 1 poor
**Rating:** 3
**Confidence:** 4

**Summary:**

This paper focuses on estimating output uncertainty caused by input uncertainty in terms of mean and variance. It utilises quasi-linear approximation to reformulate multiple activation functions in neural networks including tanh, softplus, ReLU.

**Strengths:**

The presentation is well organised and easy to follow.

**Weaknesses:**

1. Though a pipeline of uncertainty propagation from input to output is established for MLP, the relationships between the output and input uncertainties are not well established via the medium of models. It would be great if some analysis about how # layers, # nodes per layer would affect the output uncertainties are given. Such kind of insights would attract more interest of areas.

2. while it compares to a linearisation method, I am wondering if there are any other state-of-the-art methods could be compared to.

3. Most of the references are before 2015, thus I suspect whether the topic is a recent interest.

**Questions:**

see above weakness.

---

### Official Review · Reviewer_Lszd · 2023-11-01

**Soundness:** 2 fair
**Presentation:** 2 fair
**Contribution:** 2 fair
**Rating:** 5
**Confidence:** 4

**Summary:**

For safety-critical environments it is critical to asses the real-time confidence of artificial neural networks.
Considering the important target tracking domain as an example. fusing dynamical system models with large sensing networks presents a computational challenge.
The authors propose a method of a quasilinear approximation approach based on stochastic linearization to estimate uncertainty propagating through neural networks.
They also combine the method with Bayesian/Kalman Filtering. In comparison with popular approaches such as first-order (Jacobian) linearization, they showed more accurate real-time confidence estimation.

**Strengths:**

Fusing dynamical system models with large sensing networks poses  a computational challenge, but a very important domain issue.
Among many confidence estimation approaches, they propose real-time approach with better precision.
This approaches is applicable to many safety-critic applications.
Also, they tested its applicability using a variety of activation functions.

**Weaknesses:**

Their tested neural networks are very shallow and small in comparison with usual object detection networks.
Also, there exist more network structures in popular object detection networks.

**Questions:**

More experiments using popular object detection neural networks would support their claims.

---

### Official Review · Reviewer_Woe1 · 2023-11-03

**Soundness:** 3 good
**Presentation:** 3 good
**Contribution:** 1 poor
**Rating:** 3
**Confidence:** 4

**Summary:**

The authors promote the idea of using stochastic linearization (SL) to approximate nonlinear systems for analytical uncertainty estimation.  They also propose approximations of functions that are commonly used in neural networks, that enable them to perform SL analytically. They demonstrate the obtained uncertainty estimates in different contexts. Importantly, the uncertainty estimation task considered here is one where the model is known and we wish to quantify how uncertainties in the input translate into output uncertainties.

**Strengths:**

The paper is well written, and I appreciate that the authors are using classic tools like SL.

**Weaknesses:**

I have three main concerns with the manuscript:
1. Using SL to approximate nonlinear models is a well-established method, e.g., for control and nonlinear filtering. The authors have presented approximations to nonlinear functions to enable them to use SL in closed form, but novelty in those approximations is limited, especially considering that it is unclear how accurate they are.

2. The authors implicitly approximate neural networks as linear (at least locally), and even for the toy examples that they have presented it is clear that this approximation is naive. As the networks grow deeper, I am convinced that the proposed approximations will become increasingly inaccurate and, most likely, soon become of little use.  By the way, the fact that it is Gaussian densities do not remain Gaussian as we propagate the variables through a network is arguably one of the main reasons that Bayesian deep learning is so difficult.

3. Perhaps less importantly, I find the provided experiments unconvincing. It is true that SL seems to perform better than a simple Taylor expansion, but that does not imply that the presented strategy will provide accurate uncertainty estimates even in moderate sized deep neural networks.

**Questions:**

Related to my main concerns, I have two questions:
1. Can you provide a theoretical analysis of how accurate your (3) and (11) are?
2. I am not convinced that linear approximations are useful for deep networks. Can you provide any arguments for why such simple approximations are useful?

---

### Official Review · Reviewer_gBbv · 2023-11-05

**Soundness:** 3 good
**Presentation:** 2 fair
**Contribution:** 2 fair
**Rating:** 5
**Confidence:** 3

**Summary:**

In this paper, the authors present an approach that addresses the propagation of input uncertainty through neural networks. The motivation for the work stems from the practical application of target tracking. The authors introduce a novel concept based on quasilinear approximation based on stochastic linearization. To address the core challenges of uncertainty propagation, the authors derive analytical solutions for the stochastic linearization terms, encompassing both gain and bias. The paper involves derivation of the method for neural networks employing various activation functions, including tanh, sigmoid, softplus, and piece-wise linear. This comprehensive analysis reveals the versatility of their proposed approach, which can be applied to a wide range of neural network architectures. By applying their method to the two simulated examples, the authors showcase the potential impact and relevance of the proposed method.

**Strengths:**

The paper addresses a critical problem in the field of machine learning – the propagation and quantification of output uncertainty through neural networks based on input uncertainty. This is an increasingly important issue in modern machine learning, as it directly relates to the reliability and robustness of ML models. The use of the quasilinear approximation method based on stochastic linearization is not only logical but also a novel perspective. I cannot comment extensively on the literature in this field, and thus would like to read what other reviewers think about the novelty. The authors motivate the problem they aim to address by presenting a target tracking example, which demonstrates the practical implications of their research. Thus, the paper establishes its relevance and applicability. The paper also provides derivations for a series of Multi-Layer Perceptron (MLP) with common activation functions. This inclusiveness highlights the versatility of the proposed method, making it applicable to various neural network architectures. Furthermore, the authors conduct a comparative analysis against sampling methods, highlighting the advantages of their proposed approach.

**Weaknesses:**

The paper has several strengths, however there are a couple of weaknesses and questions that I have related to the paper.

1.	The authors assume that the noise on the input is always Gaussian. It would be beneficial to clarify whether this assumption holds in all practical scenarios. If it is indeed the case, it's important to discuss the limitations of this assumption. Additionally, consider elaborating on any relaxation strategies for this assumption that the authors might have explored.
2.	The proposed method approximates the output of each layer with a Gaussian distribution. This raises concerns about error accumulation, especially in large, deep networks. It would be helpful to discuss how this error accumulation scales with the depth of the network and whether the proposed method can effectively handle big networks.
3.	While the paper discusses the method's application to certain network architectures, like MLP, it is essential to address how the method can be adapted for more modern networks such as ResNet. Does the approximation apply to each ResNet block, and if so, what considerations should be taken into account for these blocks?
4.	In MLP-tanh activation function section, the tanh is also approximated, thus there are multiple sources of approximation errors. I believe a section on quantifying errors from different sources is important.
5.	The method of Wang et al. (2016) is applicable to tanh activation layers, so a comparison with this method would provide insightful context for the proposed approach. It's essential to explain any specific reasons for omitting this comparison in the current paper.
6.	The paper mentions GELU in Section 2.3 without providing any introduction.
7.	In Table 1, there is a significant difference between MC100 and MC500. It would be beneficial to explain whether MC100 involves 100 independent feedforward predictions on samples from the input distribution. If so, why is there such a notable contrast between MC100 and MC500, considering that feedforward predictions are independent and performed in batches?

**Questions:**

Please see the weakness section.

---

### Meta-Review · Area_Chair_9BFr · 2023-12-05

**Metareview:**

This paper was reviewed by four reviewers. The reviewers appreciated the topical and versatile approach. However, the reviewers also had multiple concerns. The Gaussian assumption seems to break down in deeper models, and the authors do  not thoroughly discuss error accumulation in deep networks or adaptability to modern architectures like ResNets. Additionally, the paper lacks comprehensive experimental validation and up-to-date comparisons with other methods, and could benefit from a deeper analysis of how network structure affects output uncertainties. Thus, the reviewer consensus is to reject this work in its current form.

**Justification For Why Not Higher Score:**

Even the authors seem to agree.

**Justification For Why Not Lower Score:**

N/A

---

### Decision · Program_Chairs · 2024-01-16

Reject